# Blood and Lymphatic Vasculatures On-Chip Platforms and Their Applications for Organ-Specific In Vitro Modeling

**DOI:** 10.3390/mi11020147

**Published:** 2020-01-29

**Authors:** Aria R. Henderson, Hyoann Choi, Esak Lee

**Affiliations:** 1Nancy E. and Peter C. Meinig School of Biomedical Engineering, Cornell University, Ithaca, NY 14853, USA; arh268@cornell.edu; 2Department of Biological and Environmental Engineering, Cornell University, Ithaca, NY 14853, USA; hc533@cornell.edu

**Keywords:** blood vessels, lymphatic vessels, vasculatures-on-a-chip platforms, organ specificity, in vitro models, brains, intestines, disease models-on-a-chip platforms, micro-physiological systems

## Abstract

The human circulatory system is divided into two complementary and different systems, the cardiovascular and the lymphatic system. The cardiovascular system is mainly concerned with providing nutrients to the body via blood and transporting wastes away from the tissues to be released from the body. The lymphatic system focuses on the transport of fluid, cells, and lipid from interstitial tissue spaces to lymph nodes and, ultimately, to the cardiovascular system, as well as helps coordinate interstitial fluid and lipid homeostasis and immune responses. In addition to having distinct structures from each other, each system also has organ-specific variations throughout the body and both systems play important roles in maintaining homeostasis. Dysfunction of either system leads to devastating and potentially fatal diseases, warranting accurate models of both blood and lymphatic vessels for better studies. As these models also require physiological flow (luminal and interstitial), extracellular matrix conditions, dimensionality, chemotactic biochemical gradient, and stiffness, to better reflect in vivo, three dimensional (3D) microfluidic (on-a-chip) devices are promising platforms to model human physiology and pathology. In this review, we discuss the heterogeneity of both blood and lymphatic vessels, as well as current in vitro models. We, then, explore the organ-specific features of each system with examples in the gut and the brain and the implications of dysfunction of either vasculature in these organs. We close the review with discussions on current in vitro models for specific diseases with an emphasis on on-chip techniques.

## 1. Introduction

The circulatory network of the human body is composed of the cardiovascular system and the lymphatic system that carry blood and lymph, respectively. The compositions of these circulating liquids are different in blood and lymphatic vessels, and they provide important information about the physiological processes. The endothelial cells (ECs) derived from the mesoderm differentiate into distinct lineages that produce the blood vessels and the lymphatic vessels separately. As a result, blood and lymphatic endothelial cells have different transcriptional features. To complicate matters, these cells also exhibit organ-specific and disease-specific transcriptomes. As a result, disease conditions alter the composition of blood and lymph affecting the vessel integrity, and vasculature dysfunction is implicated in various diseases such as atherosclerosis and lymphedema.

The importance of understanding the key roles of the vasculatures in diseases have led to great efforts in developing relevant in vitro models. In order to recapitulate the dynamic flows of the circulating liquids and the tissue–tissue interactions between the vasculatures and the organ tissues, microfluidic devices, termed on-chip devices, have become an attractive in vitro modeling platform. As a result, many vasculature on-chip models have been created and used to study the roles of the circulatory system especially in inflammation, cancer metastasis, and drug delivery. However, vasculature-on-chips still receive less attention as compared with other organ-on-chips. As vasculatures are integral components that mediate the functions of other organs, the development of vasculature-on-chips should advance our understanding of vascular diseases and systemic responses.

In this review, we discuss the physiology of the lymphatic and blood vascular systems and the application of organ-on-a-chip technology in studying these systems. We further highlight the gut and the brain for their vasculature system physiology and the current status for applying microfluidic chip technology to study the vascular functions in these organs. We then analyze the current understanding in vasculatures and chip technologies to discuss key considerations in designing future vasculature-on-chips. We envision that these chips could ultimately be included in a whole-circulation-on-a-chip or in a body-on-a-chip, which would allow one to study organ–organ interactions, pharmacokinetics, toxicology, and pharmacodynamics. The collective understanding of the organ functions would enable the development of more effective therapeutics that simultaneously target various physiological processes related to multifactorial diseases such as diabetes, cancer, and neurodegenerative diseases.

## 2. Blood Vasculature

As discussed above, a healthy human body is supported and maintained through two major circulatory systems, the cardiovascular system and the lymphatic system. Powered by the heart, the cardiovascular system is a powerful and complex network necessary to sustain health. The cardiovascular system uses blood vessels, and similar to lymphatic vessels, consist of a single layer of ECs. In the cardiovascular system, arteries, veins, and capillaries work together with the heart to deliver nutrients and paracrine signaling molecules, transport cells, and metabolites, and promote gas exchange throughout the body [1]. Complementarily, the lymphatic system is an open circulatory system that collects interstitial fluid, including metabolic wastes, and channels them into venules. Through these venules, this fluid continues in a closed vascular circulatory system as blood, where it reaches organs such as the lungs and the kidneys that reoxygenate the blood and filter wastes from it, respectively [2].

### 2.1. Development and Heterogeneity of Blood Vessels

Blood vessel development begins with primordial ECs differentiating from the mesodermal cell line in embryos. This occurs primarily through bone morphogenetic protein 4 (BMP4), fibroblast growth factor 2 (FGF2), and Indian hedgehog (IHH) signaling. BMP4 has been shown to be at the top of the signaling chain, causing a FGF2-dependent sequence via fibroblast growth factor receptor 1 (FGFR1) to induce and regulate the pattern of the mesoderm. As a matter of fact, BMP4 has also been shown to be sufficient in vitro to induce differentiation of the mesoderm into primordial ECs. In contrast, IHH is produced by the visceral endoderm in vivo but its role is less specifically understood than that of BMP4 or FGF2; however, as embryos deficient for IHH and its receptor display some, but subnormal, EC differentiation, IHH likely plays a complex, underappreciated role in EC development [3]. From primordial ECs, vascular endothelial growth factor (VEGF) plays a role in regulating vasculogenesis, a process of angioblasts differentiating into ECs de novo. Through vascular endothelial growth factor receptor 1 and 2 (VEGFR1 and VEGFR2), vascular endothelial growth factor A (VEGF-A) is involved in EC propagation and survival and interacts with the neuropilin 1 and 2 co-receptors (Nrp-1/2). This propagation leads to the formation of angioblasts, while retinoic acid (RA) signaling leads to the formation of hemogenic ECs. Hemogenic ECs undergo further differentiation via Notch signaling and *Runx1* (also known as *AML1*) expression to generate hematopoietic progenitor cells. These two cell types, i.e., hematopoietic progenitor cells and angioblasts initiate the formation of the cardiovascular system. Vasculogenesis begins when growth factors and morphogens lead to the expression of ephrin B2 in arterial-fated ECs and ephrin type B receptor 4 (EphB4) in venous-fated ECs, which segregate from a common precursor vessel. In addition to differences in gene expression, higher levels of VEGF-A induce arterial ECs while low-to-intermediate levels are indicative of venous ECs. Chicken ovalbumin upstream promoter-transcription factor II (COUP-TFII) deletion contributes to the arterialization of veins, and ectopic expression of COUP-TFII results in artery and vein fusion [3]. As shown in Figure 1b, lymphatic ECs stem from embryonic venous ECs through further specification. This specification is largely mediated by prospero homeodomain transcription factor 1 (*Prox1*) expression, which is co-regulated by COUP-TFII and Sex-determining region Y (SRY)-box Transcription Factor 18 (*Sox18)* [4].

In both embryos and adults, angiogenesis is the process of forming new blood vessels from quiescent cells already present in the network. Angiogenesis can be described as either sprouting or intussusceptive angiogenesis. Sprouting angiogenesis involves vessels sprouting or branching from endothelial cells in current vessels as directed by growth factors such as VEGF-A. It has been shown that a lack of available nutrients and oxygen for tissue is the primary stimulus for blood vessel formation in this fashion [1]. This partially explains why rapid angiogenesis is seen in tumors, as their local environments require ample metabolic resources to support their uncontrolled growth. Sprouting angiogenesis is typically initiated by leading ECs (or “tip cells”) within the current vasculature but can also face angiogenic competition from the following “stalk cells.” Tip cells are the migratory and invasive cells found at the end of blood vessels and are believed to be the guiding cells for new sprouts. Using their filopodia for better migration, tip cells can sense proangiogenic VEGF-A gradients through their high expression levels of VEGFR2 and VEGFR3, which lead to downstream expression of delta-like protein 4 (DLL4). In turn, DLL4 activates Notch signaling. This is paired with activin receptor-like kinase (ALK) signaling, and the signal combination prevents the trailing stalk cells from becoming tip cells and establishes their stalk-like behavior of elongation, proliferation, vessel stability, and eventual lumen formation [5].

Alternatively, intussusceptive or splitting angiogenesis relies on restructuring or duplicating existing blood vessels rather than extending current blood vessels. Intussusceptive angiogenesis begins with the formation of a “pillar” or “post.” From the exterior of the vessel, these pillars appear as small, nonrandom holes in the endothelium; however, these pillars are EC transluminal microstructures and are often difficult to identify and quantify using traditional light microscopy methods. Upon formation, these pillars have an extracellular matrix deposited by invading mural cells (pericytes and fibroblasts) and gradually increase in size until they split the vessel. While similar structures exist in fish gills, mollusks, and crustaceans, these specific structures are unique to mammalian anatomy. In both cases, they are important features for optimizing bulk fluid transport by providing increased opportunity for the exchange of soluble factors and progenitor cells among other blood-borne elements. When the pillars are used in intussusceptive angiogenesis, they can modify the blood vessels in the following three different ways: (1) by modifying the branching angle of splitting vessels, (2) by duplicating current vessels, and (3) by reducing the usage of energetically-inefficient or redundant vessels [6]. Intussusceptive angiogenesis is driven by a combination of blood flow, VEGF signaling, and Notch1 activation. Increased blood flow has been shown to increase the incidence of intussusceptive angiogenesis in developmental angiogenesis in healthy tissue as well as pathological angiogenesis, such as that of tumor vasculature [7]. The exact molecular mechanisms behind intussusceptive angiogenesis are still largely unknown, but there has been evidence that an all-stalk phenotype could be the first step in generating a pillar. If this is the case, intussusceptive angiogenesis would be promoted by high VEGF concentrations and complete Notch1 activation in the ECs [8].

### 2.2. Structure and Functions of Blood Vessels

The vasculature of the cardiovascular system can be divided into three major categories based on function and structure. Arteries and arterioles bring high-pressure blood from the heart to other organs in the body, veins and venules bring low-pressure blood back to the heart using semilunar valves to prevent backflow, and capillaries act as the interconnecting network between arterioles and venules. With a diameter of 5 to 10 µm, capillaries are the smallest vessels among these and exchange their contents with the tissues they directly contact. All three categories of vessels have the same basic structure, i.e., a single layer of ECs, called the endothelium, covered with a basement membrane and, then, by mural cells, but the specifics of each of these layers vary with vessel function. Hence, the endothelium can be continuous, fenestrated, or discontinuous [1]. A continuous endothelium acts as strong barrier and is especially seen in capillaries of the blood–brain barrier (BBB), while a fenestrated endothelium contains many pores and can be found in the glomeruli of kidneys, where these pores facilitate the diffusion of water, as well as in the intestinal mucosa and the glands for both endocrines and exocrines. A discontinuous endothelium contains an incomplete single layer of endothelial cells and easily allows the passage of small molecules, which is particularly useful in the filtration system of the liver and in the sinusoidal vasculature of the bone marrow [1]. Beyond these different types of endothelium, blood vessels also vary greatly in their mural cell content. Mural cells are generally categorized into either vascular smooth muscle cells (vSMCs) or pericytes [9]. The vSMCs play an important role in maintaining the pressure, tone, and stability in blood vessels, and therefore are found in especially high abundance on the outside of arteries and arterioles, as these blood vessels contain high-pressure blood from the heart and must maintain this pressure to propel the blood through the entire vascular system. Pericytes are responsible for the transport and barrier properties of blood vessels through signaling and have been shown to have contractile properties through their cytoplasmic extensions that wrap around the basement membrane and endothelium [10]. For their contractile and barrier functions, these cells are typically found in great numbers around veins, venules, and capillaries; however, it should be noted that both pericytes and vSMCs can be found surrounding the same type of blood vessel, such as in large veins, where pericytes and vSMCs work together with semilunar valves to facilitate smooth, unidirectional blood flow [11]. These characteristics are diagrammatically represented in Figure 1a.

### 2.3. In Vitro Models of Blood Vessels

For modeling blood vessels in vitro, human umbilical vein endothelial cells (HUVECs) are a popular choice due to their high availability as compared with other types of blood endothelial cells (BECs). HUVECs can be from primary cells or as part of an immortalized fused cell line. Primary HUVECs can be purchased commercially or isolated and maintained with relative ease. Regardless of source, HUVECs are also known for displaying several different endothelial markers, such as PECAM-1 (platelet endothelial cell adhesion molecule-1), VCAM-1 (vascular cell adhesion molecule-1), and ICAM-1 (intercellular adhesion molecule-1) [12]. Two-dimensional (2D) monolayers, three-dimensional (3D) organoids, and 3D co-culture models have all been created using HUVEC, microvascular ECs, and co-cultures techniques that combine ECs with other cell types for specific physiological investigations.

In general, 2D monolayers are important for drug testing and the early stages of permeability testing, whereas 3D options capture the nuances of in vivo microenvironments more accurately, including important structures in the extracellular matrix (ECM). Tissue-engineered blood vessels (TEBVs) were 3D models that have been utilized not only as prospective graft and bypass materials but also as disease models for atherosclerosis in pharmacological studies [13]. In diabetic vasculopathy research, CD31 (cluster of differentiation 31, also known as PECAM-1: Platelet endothelial cell adhesion molecule-1) expressing induced pluripotent stem cell derived ECs (iPSC-Ecs) were cultured with pericytes to create vascular organoids. These organoids expressed the pericyte coverage and matrix deposition seen on Ecs in vivo and, upon in vivo exposure to mice with a diabetic milieu, displayed microvasculature changes seen in diabetic patients [14]. Some models even use 3D bioprinting to recreate vessels with a continuous endothelium and surrounding layers of smooth muscle cells and fibroblasts in a collagenous matrix. These printed vessels were cultivated in dynamic, fluidic biochambers, and had high cell viability, as well as detectable levels of VE-cadherin, smooth muscle actin, and type IV collagen after weeks of culture [15].

Across all areas of vascular research, flow is required for a functional model, and microfluidics options for blood vessel models have also increased in number recently. V. van Duinen et al. used 3D collagen-I matrices in 96-well microtiter plates to create 3D vessel structures and mimicked flow using a rocker to investigate the relationship of the permeability of microvesesels and signaling molecules such as VEGF, tumor necrosis factor alpha (TNFα), and other cytokines [16].

A promising in vitro model technique that incorporates physiological flow while allowing for specific 3D structures is the use of microfluidic chips (organs-on-chips). These devices use a variety of cell types and sources, including primary endothelial cells, primary pericytes, and even induced pluripotent stem cell (iPSC)-Ecs [17,18,19,20,21]. A multiwell vascular network tissue platform has been created that is perfusable (see Figure 2a). This platform is ideal for drug screening as proven by the ability of 70 kDa fluorescein isothiocyanate (FITC)-dextran to flow through the vascular network (see (i) in Figure 2a). These vascular networks also showed claudin-5 expression (see (ii) in Figure 2a) and vascular endothelial (VE)-cadherin expression (see (iii) in Figure 2a) both of which are integral for proper blood endothelium barrier function, making this chip physiological in terms of barrier permeability [17]. Other variations use iPSC-Ecs to create functional microvessels complete with fully formed lumen and laminin deposited in the basement membrane (see (i) and (ii) in Figure 2b). Using a line of iPSC-Ecs would provide the advantages of reducing the donor-to-donor variability seen in primary cell lines and specific gene editing to create pathology- and possibly patient-specific gene expression [18]. Combined blood vessel and lymphatic vessel chips (see (i) in Figure 2c) provide a unique insight into interactions between the two microvessel types and can even be used to investigate the effects of growth factors such as VEGF-C on both angiogenesis and lymphangiogenesis simultaneously (see (ii) in Figure 2c) [19]. Some chips choose to use multiple cell types in an attempt to better recapitulate the vascular and perivascular components of blood vessels, such as including pericytes seeded adjacent to endothelial cells (see (i) and (ii) in Figure 2d). This multicomponent approach allows one to form a layered architecture that matches what is seen in human physiology (see (iii) in Figure 2d) [20]. Across all on-chip models, the key features are the ability to provide continuous physiological flow and precisely control the input and output components. As seen in Figure 2e, static conditions as compared with flow conditions affect the structure of the microvessels, and the input of the fluid that flows through the vessel also plays a major role in shaping the microvessel [21]. Using different aspects of these on-chip models promises the potential of creating one comprehensive blood-vessel-on-a-chip or the possibility of incorporating additional components to create vasculatures-on-chips for specific organs.

## 3. Lymphatic Vasculature

### 3.1. Development and Heterogeneity of Lymphatic Vessels

While many hypotheses regarding the developmental origin of lymphatic vessels have been suggested, studies in the past years have led to a general agreement among researchers on the ”centrifugal” theory. According to this theory, the lymphatic vessels in mammals emerge from the embryonic vein after the establishment of the cardiovascular system [2]. Subsequent studies have identified several regulators in lymphatic endothelial cell (LEC) development and specification. During embryogenesis, the mesoderm differentiates from embryonic stem cells to give rise to endothelial cell precursors, angioblasts. Primitive vasculature is formed from the endothelial cells (ECs) differentiated from these angioblasts [22]. While Notch signaling induces arterial specification of ECs, COUP-TFII inhibits the Notch signaling to drive venous specification [22]. Further LEC commitment is modulated by prospero homeobox transcription factor 1 (PROX1). SOX18, which is activated by mitogen-activated protein kinase/extracellular signal-regulated kinase (MAPK/ERK) signaling [23], induces PROX1 expression in LEC precursors among the venous ECs highly expressing LYVE-1 (lymphatic vessel hyaluronan receptor-1), which serves as a definitive lymphatic endothelial marker. PROX1 further dimerizes with COUP-TFII to induce the expression of major genes for lymphangiogenesis, including vascular endothelial growth factor receptor 3 (VEGFR-3), podoplanin (PDPN), and chemokine (C-C motif) ligand 21 (CCL21) [24,25]. The committed LECs bud out from the cardinal vein in response to mesenchymal vascular endothelial growth factor C (VEGF-C) signals, which activate VEGFR-3. The following lymph sac formation from these cells migrating away from the cardinal vein is mediated by PDPN. By activating C-type lectin receptor 2 (CLEC-2), PDPN induces platelet aggregation that requires SH2 domain containing leukocyte protein of 76kDa (SLP-76), spleen associated tyrosine kinase (SYK), and phospholipase C gamma 2 (PLCG2) activities to separate lymphatic vessels from blood vessels [26,27]. As the primitive lymphatic vessels mature, the lymphatic vascular network expands with lymphatic capillaries sprouting out from the nascent lymphatic plexus, of which the rest remodels into complete collecting lymphatic vessels. The maturation process of lymphatic vessels is mediated by several regulators, including EphrinB2, adrenomedullin (AM), and angiopoietin 2 (Ang2) [28]. Importantly, the nuclear factor of activated T cells (NFATc)-1 and forkhead box C2 (FOXC2) mediate the differentiation of collecting lymphatic vessels from capillaries by controlling genes for valve formation [29]. Furthermore, platelet-derived growth factor B (PDGFB) controls lymphatic smooth muscle cell recruitment specific to the collecting lymphatic vessels during the remodeling process [30].

Although this vein-derived origin of lymphatic vasculature was believed to be the only origin of lymphatic vessels for many organs, evidence of non-venous origins of lymphatic vasculature across different tissues have also been reported. For example, the dermal lymphatic vessels in the lumbar and dorsal midline skin were found to be derived from non-Tie2-lineage cells, which are not derived from veins [31]. Additionally, cardiac lymphatics were also reported to have a non-venous developmental origin to become the putative hemogenic endothelium [32]. Such organ-specific differences in the non-venous development of lymphatics can be attributed to diverse microenvironments in different organs that serve the tissues’ unique functions. Furthermore, lymphatic vessels can also be derived from hematopoietic stem cells [33] and several attempts at lymphatic regeneration during pathological conditions have been successfully achieved by the differentiation of LECs from pluripotent or multipotent stem cells [34,35].

Organ-dependent characteristics of lymphatic vessels extend even beyond the developmental origin. Different molecular regulators can drive organ specification of LECs, although most LECs in different organs are represented by the same LEC markers including PROX1, VEGFR3, PDPN, lymphatic vessel endothelial hyaluronan receptor 1 (LYVE1), CCL21, and integrin alpha-9 (ITGA9) [36]. For example, the transcription factor GATA binding protein 4 (GATA4) has been found to induce liver sinusoidal endothelial cell specification, which is integral for hepatic microvasculature integrity [37]. Correspondingly, lymphatic functions and implications in health and disease also vary by organ [36]. However, organ-specific molecular signatures of lymphatic vessels largely remain unknown despite the existing evidence of lymphatic heterogeneity.

### 3.2. Structure and Functions of Lymphatic Vessels

Spread throughout the body, the lymphatic vasculature is a blind-ended network that transports the interstitial fluid, i.e., lymph, one-way (as compared with the blood vasculature, which is a continuous circulatory loop) [28]. The lymphatic vasculature was previously thought to be present in all tissues except for the brain, and the presence of lymphatics there had been ambiguous until some lymphatic vessels were recently discovered in the central nervous system [38,39].

Lymphatic vasculature is composed of lymphatic capillaries (or initial lymphatic vessels) and collecting lymphatic vessels that connect lymph nodes and transport lymph collected from the interstitial space of peripheral tissues. When blood oxygenates a tissue, the high blood pressure in the capillaries forces the release of plasma into the interstitial space of the tissue. While much of the leaked plasma, now called interstitial fluid, is reabsorbed by the vascular capillaries, a small amount of the fluid remains in the interstitial space. In healthy conditions, lymphatic capillaries take up the remaining fluid, which becomes lymph, to maintain tissue fluid homeostasis. Most of the collected lymph is returned to the blood vasculature via entry into the thoracic duct, which is located at the junction of the left jugular and subclavian veins. However, some lymph from the right side of the body above the diaphragm is collected by the right lymphatic duct, which is located at the junction of the right jugular and subclavian veins, also to be returned to the bloodstream [40].

While both the lymphatic capillaries and collecting vessels are lined by LECs in monolayer, they have distinct structures to serve their different functions. The blind-ended lymphatic capillaries have some flap-like minivalves between adjacent lymphatic endothelial cells to allow the uptake of molecules and immune cells, as driven by the interstitial pressure gradient. These lymphatic capillaries have discontinuous button junctions and basement membranes that contribute to some degree of permeability for the uptake of lymph components. In contrast, collecting lymphatic vessels have continuous zipper junctions and basement membranes that make them less permeable to support their main function of lymph transport (rather than uptake). Moreover, the outer side of these collecting vessels is lined by lymphatic smooth muscle cells, which are not present in the lymphatic capillaries, to help pump lymph. Furthermore, the collecting vessels possess valves that are positioned on the luminal side in a way to prevent the backflow of lymph, unlike the capillaries [41].

### 3.3. Immune Regulation by Lymphatic Vessels

In addition to the maintenance of tissue fluid homeostasis, lymphatics play other important roles in health and disease. As reflected in the variety of components of lymph that include lipids, proteins, fat-soluble vitamins, exosomes, and immune cells [42,43], one significant function of lymphatic vasculature is immune regulation. By transporting antigens, microbes, danger signals, cytokines, and immune cells, lymphatic vessels mediate immune responses both locally and systemically. The interstitial fluid drained from the peripheral tissues contains foreign antigens, tissue-specific self-antigens, and local signals that inform about the health of the peripheral tissue. When taken up by the blunt-ended lymphatic capillaries, the lymph flow conveys these soluble factors to lymph nodes, where populations of immune cells reside. The conveyed signals trigger either stimulatory or tolerogenic immune responses in lymph nodes [43]. Moreover, lymphatic vessels actively mediate recruitment, uptake, and migration of immune cells to lymph nodes by expressing various chemokines and adhesion molecules. The most well-known example is the expression of CCL21, which is the ligand for the C-C chemokine receptor type 7 (CCR7) on dendritic cells (DCs) and T cells. The CCL21/CCR7 signaling is responsible for immune cell trafficking via lymphatics in conjunction with expression of the ICAM-1 adhesion molecule by LECs. These provide the biochemical signals for immune cell recruitment and migration [44]. More interestingly, a recent study found that LYVE-1, which is neither a chemokine nor an adhesion molecule, also mediates the entry of DCs into the lymphatic vessels by recognizing hyaluronan on the surface of DCs [45].

The homing immune cells loaded with periphery-derived antigens induce maturation of lymph node-resident immune cells. When the naive immune cells in lymph nodes are activated by soluble antigens or homing immune cells delivered by lymph, these activated or tolerized cells can return to the original tissue or disseminate to other tissues for local and systemic immune responses [43,44]. Furthermore, the lymphatic smooth muscle cell-mediated motility of colleting lymphatic vessels allows temporal delivery of antigens and immune cells to lymph nodes [46]. In a murine model of collecting lymphatic contraction, this autonomous movement of lymphatic vessels was found to impact suppressive immune responses [47]. Since LECs in lymph nodes have been reported to further actively regulate immune responses by other mechanisms such as expression of major histocompatibility complex (MHC) molecules [48,49], whether the LECs comprising the lymphatic vessels could also have other immuno-modulatory mechanisms is a study topic of interest. Diverse roles of lymphatics in regulating immunity have been widely studied, and more comprehensive details can be found elsewhere [43].

### 3.4. In Vitro Models of Lymphatic Vessels

Although less studied than blood vasculature, lymphatic vasculature has been an attractive area of research to many researchers due to its implications in numerous diseases. Lymphatic vasculature dysfunction has been found to be involved in inflammatory diseases, cardiovascular diseases, obesity, and, most extensively, lymphedema and cancer metastasis [50,51]. In these diseases, pathological lymphangiogenesis and lymphatic remodeling have been commonly observed [40,47,48,49,50]. The surrounding cells, including stromal and immune cells, constitute a certain microenvironment that induces the changes in the lymphatic vessels by secreting biochemical factors. For example, during cancer, lymphangiogenic factors in the tumor microenvironment secreted by tumors and other surrounding cells remodel lymphatic vessels, facilitating cancer progression and metastasis [52,53].

The extensive involvement of lymphatic vasculature in several diseases has led to several attempts to recapitulate lymphatics in 2D and 3D in vitro models. LECs used were purchased from several vendors but, they could also be isolated from humans or mice in house. Furthermore, LECs differentiated from stem cells were also used for in vitro lymphatic vessel models. The LECs were, then, cultured on plastic or matrix-coated plates or Transwell inserts. Incorporation of biochemical factors or other cell types helped, but were not required, to form the microenvironment that leads to lymphatic microvascular networks. Moreover, 3D environments could be created in spheroids or thicker matrices. Furthermore, exerting flows using flow chambers or simple rocker has been found to be critical to recapitulate physiologically similar lymphatic formation and functions [54]. These previous in vitro models have advanced our understanding of the effects of various physiological factors on the properties of lymphatics. Interestingly, the emerging microfluidic in vitro system, an organ-on-a-chip, has shown great promise in modeling the dynamic microenvironment surrounding the lymphatic vessels on a micro scale. Gong et al. modeled a tubular lymphatic vessel in a microchannel embedded in a collagen gel chamber (see Figure 3a). The microscale fluidic system was used to characterize the lymphatic barrier function and to model tumor microenvironments surrounding the lymphatic vessels. The study showed the usefulness of the microphysical system as disease models for controlled mechanistic studies [55]. The blood-and-lymphatic-vessels-on -a-chip was also enabled by the bioprinting technology. Instead of needles or rods to make microchannels, Zhang et al. utilized a nozzle to fabricate hollow tubes (see Figure 3b). By adjusting the bioink flow rates, this bioprinting method enables independent control of the wall thicknesses of the tubes. More interestingly, the authors also recapitulated the one end-blinded characteristic of lymphatic capillaries using this method [56]. In addition to the luminal flow through the channels, Kim et al. utilized a microfluidic platform to generate the interstitial flow across a channel. The authors investigated the effects of interstitial flow on the lymphatic sprouting during lymphangiogenesis. Two fluidic channels that separate the central lymphatic channel from the two fibroblast channels on the sides were used to control for the interstitial flow pressure gradient across lymphatics and the concentration of biochemical factors (see Figure 3c). The authors elucidated the effect of interstitial flow on the lymphatic filopodia projections and the guided lymphatic growth in the presence of biochemical stimulants [57]. Although these microfluidic chip models are powerful tools to study lymphatic vessels as described above, they can be further tailored to recapitulate more physiologically similar structures of the lymphatic vessels. Incorporation of more biological components that constitute the microenvironments surrounding the lymphatic vessels can further advance the understanding of the tissue–tissue interactions and the effects of diseased conditions.

## 4. Gut Microenvironment and In Vitro Models

### 4.1. Structure and Functions of the Intestines

The intestine is largely divided into the small and large intestines, which differ in function. Finger-like projections of the epithelium called villi contribute to a more optimal nutrient absorption function in the small intestine by providing a larger surface area. The villi lengths decrease from the duodenum to the ileum, reflecting greater nutrient absorption at the duodenum. The large intestine, however, lacks the villi structure and absorbs less nutrients than the small intestine; instead, it is a major water absorption site that creates feces from the chyme generated in the small intestine. It also harbors the largest number of microbes, which play important roles in many physiological processes including metabolism and immunity [58,59].

Starting from the luminal side, the intestinal layers are comprised of the mucosa, submucosa, muscularis, and serosa. The mucosa, composed of the epithelium, the lamina propria, and the muscularis mucosae, supports most of the physiological functions of the intestine, including nutrient absorption and immune regulation [59,60].

The intestinal epithelial cells (IECs) regulate immune responses by acting as a barrier against external environmental stresses and directly interacting with the immune cells in the lamina propria. IECs secrete antimicrobial peptides (AMPs) and express pattern-recognition receptors (PRRs) including toll-like receptors (TLRs) that recognize the distinct patterns of the microbes, called pathogen-associated molecular patterns (PAMPs). Upon TLR activation, the IECs secrete cytokines that activate immune cells including dendritic cells, macrophages, T cells, and B cells. IECs also act as antigen presenting cells by expressing MHC molecules that relay the microbial signals in the lumen to the immune cells that reside in the lamina propria. The activated immune cells are then transported to the lymph nodes to propagate the immune responses [60,61].

These roles of IECs as mediators of the intestinal immune responses are important for maintaining the homeostasis with the gut microbiome. The epithelium senses the microbial activity and regulates the entry of the microbes and the microbial products into the body [60,61]. The intestinal epithelium plays an important role in keeping the immune balance with the gut microbiome by tolerizing commensal microbes while eliminating pathogenic microbes [60,62].

Corresponding to the constant exposure to environmental stresses, intestinal epithelial cells have a high turnover rate. As the epithelial cells on the tip of the villus constantly undergo apoptosis, the intestinal stem cells located in the invaginations, called crypts, rapidly proliferate and migrate up the crypt-villi axis to differentiate and replace the shed epithelial cells [63].

### 4.2. Intestinal Vasculature

#### 4.2.1. Gut Blood Vessels

Below the intestinal epithelium and the basement membrane, the vascular, immune, and nervous systems form a dense network to maintain and mediate the functions of the intestine. Blood vessels, lymphatics, and nerves extend throughout the loose connective tissue called the submucosa [64]. While the nerves in the submucosa are important for regulating the intestinal metabolic processes [65], the vascular systems ensure sufficient oxygen supply to the organ and mediate systemic responses to the environmental changes in the gut [66,67,68].

In the intestine, three main arteries supply blood to different parts of the intestinal tract. The topmost celiac artery supplies blood to the stomach and the duodenum. Branching out from the abdominal aorta below the celiac artery, the superior mesenteric artery supplies blood to the jejunum, ileum, ascending, and transverse colon. Lastly, the inferior mesenteric arteries support the descending colon [69].

The blood arterioles sprouting from submucosal arteries form extensive capillary networks in the mucosa. At the tips of the microcirculation networks, the deoxygenated blood flows through the venules that extend from the tip to the submucosa to join the vein. The countercurrent arrangement of arterioles and venules in the villi (see Figure 4) creates the oxygen gradient from the base (where the stem cells are) to the tip near the epithelium. The venules from the small intestine and the colon exit the intestinal layers to join the veins. The small intestine is drained by the superior mesenteric vein, whereas the colon is drained by the inferior mesenteric vein [70,71].

The blood pressure in the intestine is maintained by several mechanisms to meet the oxygen and molecule transport needs depending on different environmental stimuli. The intrinsic regulation of intestinal blood flow involves metabolic, myogenic, and flow-dependent mechanisms. The change in the interstitial oxygen gradient depends on the metabolic rates of the surrounding cells and the tissues, and can alter the intestinal blood flow [72,73,74]. The metabolism end products, such as proteins, nitric oxide, and adenosine, can also be vasoactive molecules that regulate vasodilation in response to the metabolic activity changes [66,68,75]. The myogenic autoregulation is mediated by the smooth muscle cells that line the blood vessels. The stretch of the blood vessels due to the increased transmural pressure signals the smooth muscle cells to contract [66,76].

The intestinal blood flow is further regulated extrinsically by the neurons, the gut hormones and peptides, and the absorbed nutrients. The enteric neurons that form the submucous plexus sense the environment in the lumen and regulate the blood flow by direct effects on the arterioles in the submucosa. The extrinsic sympathetic and parasympathetic nerves innervate the serosa and contribute to intestinal blood flow regulation by interacting with the enteric neurons in the submucosa [66,77]. The gut-derived molecules, including the gastrointestinal hormones and absorbed nutrients, add another mechanism to blood flow regulation by directly acting as vasodilation or vasoconstriction inducers [78].

#### 4.2.2. Gut Lymphatic Vessels

Composed of blind-end lymphatic capillaries and collecting vessels, the intestinal lymphatic vasculature returns lymph to the blood circulation via the thoracic duct. Its integrity is important to maintain the interstitial fluid homeostasis below the intestinal epithelium [67]. The lymphatic vasculature further regulates the immune system in the intestines, as it functions as a pathway for the immune cells to migrate between the intestine and the mesenteric lymph node (MLN). For example, similar to other organs, the tissue-resident DCs in the intestinal lamina propria interact with the lymphatic capillaries to migrate from the intestine to the MLN [79]. In the skin, LEC-generated CCL21 chemokine gradient mediates the migration of dermal CCR7+ DCs [80]. Similarly, the intestinal CCR7^+^ DCs follow the CCL21 gradient generated by the LECs of the intestinal lymphatics to migrate to the MLN [81]. These lymphatic trafficking DCs play a major role in immune responses by sampling the antigens from the intestine and present them to the naïve immune cells in the lymph nodes. The interactions between the antigen presenting DCs and the T cells mediate the immune tolerance to the food molecules or the gut microbes [81,82]. Further investigation into the types of immune cells that migrate in the intestinal lymphatics and how those lymphatic trafficking immune cells determine the systemic immune responses is needed to better understand the various inflammatory and autoimmune diseases affected by the intestine-borne antigens.

In the intestine, the lymphatic capillaries are present both in the mucosa and the submucosa. The submucosal lymphatic capillaries run parallel to the intestinal wall along with the arteries and the veins. In the mucosa, the lymphatic capillaries sprout from these submucosal lymphatic capillaries perpendicularly, terminating with blind ends. The lymphatic capillaries in the center of the villi of the small intestine are called lacteals. These are surrounded by the blood microcirculation and stromal cells [83,84]. A similar structure can also be found between the crypts in the mucosa of the colon [85].

The LECs comprising the intestinal lymphatic vessels have zipper-like cell–cell junctions and intraluminal valves as in other organs. These features help the vessels serve their major function of lymph transport. The submucosal lymphatic capillaries also have similar structure to the lymphatic capillaries in other organs. The LECs comprising the submucosa lymphatic capillaries form button-like junctions and are not surrounded by mural cells [67]. However, the lacteals in the mucosa have some distinct features as compared with the lymphatic capillaries in other organs. There are two types of junctions in the lacteals (see Figure 4). While button-like junctions connect the LECs in the stalk region, zipper-like junctions are observed at the tips of the lacteals. Furthermore, the lacteals have filopodia that help scrutinize the surrounding microenvironment. These protrusions allow the lacteals to respond to the immunogenic and dietary signals that enter the lamina propria from the intestinal lumen [83]. Interestingly, they exhibit contractile motions despite the absence of the smooth muscle cells (SMCs) directly surrounding the lacteals. The peristaltic motion to absorb and transport lymph is attributed to the villi SMCs movement controlled by the autonomic nervous system [86]. The SMCs in the villi also affect the transcriptome of the lacteal LECs by producing VEGF-C, which regulates notch ligand delta-like protein 4 (DLL4) expression in lacteal LECs [83,87].

Furthermore, adult lacteal LECs undergo continuous remodeling with higher proliferation rates than the LECs in the submucosa and other organs that mostly remain at rest [83]. This higher proliferative capacity could be important for maintaining the lacteal integrity. As states above, the epithelial cells at the tips of the villi are constantly replaced by new epithelial cells pushed up due to the exposure to many environmental stresses present in the lumen of the intestine, and these cells are differentiated from the stem cells at the crypts. Similarly, the damaged LECs at the tips of the lacteals exposed to constant environmental stresses would also be replaced by new LECs owing to high proliferation [63]. The proliferating LECs can further set the lacteals in the state prepared for lymphangiogenesis to compensate for the changes in pathological conditions.

Adding to the structural differences, lacteals have another function of absorbing the dietary fat absorbed by the enterocytes (which processes them into chylomicrons) [88]. In addition, other dietary molecules, such as fat-soluble vitamins and cholesterol, are also incorporated in the chylomicrons to be transported by the intestinal lymphatic vessels [89,90]. Despite these distinct features of the intestinal lymphatic vasculature as compared with those of the lymphatic vasculature in other organs, the intestinal lymphatic endothelial cells’ transcriptome, phenotypes, and functions have not yet been fully characterized.

#### 4.2.3. Intestinal Vasculature in Diseases

The mucosal microenvironment in the intestine is comprised of various factors that cannot be found in the interstitial spaces of other organs. In addition to the extracellular matrix, stromal cells, immune cells, dietary molecules, gut microbial metabolites, and invading microbes are all present in the mucosal microenvironment of the intestinal vasculature. This dynamic microenvironment poses a variety of environmental stresses to which the intestinal vascular cells must adjust. The blood and lymphatic vessels should be continuously remodeled in order to compensate for any unfavorable shifts in the intestinal environment. This suggests that the vasculature integrity is important for the homeostasis of other cells in the intestine. In turn, the activities of the cells in the lamina propria affect the intestinal vasculature by regulating angiogenesis and lymphangiogenesis factors [87,91].

Intestinal ischemia is a condition of abnormal blood flow to the intestine due to blockage of blood vessels. When the blood flow to the intestine decreases, intrinsic vasoregulatory mechanisms dilate arterioles and recruit capillaries. Extensive collateral networks are also formed to compensate for the decreased oxygen delivery due to the decrease in blood flow [91,92]. In addition to the structural remodeling, the permeability and hydraulic conductivity of the vessels increase during ischemia [93,94]. Structural and functional alterations of the intestinal vasculatures can also be observed in various gastrointestinal diseases. For example, in inflammatory bowel disease (IBD), the inflammatory condition is regulated by the colonic vasculatures, and increased angiogenesis and lymphangiogenesis have been reported in a number of studies [95,96,97,98]. However, whether the increased vessel densities resolve or sustain the IBD is still unclear, these new vasculatures are leaky, which can exacerbate the inflammatory condition. Increased vascular permeability is another key feature during inflammation [99,100]. Additionally, stage-dependent changes in the blood flow have been reported in experimental colitis models. Blood flow rates increased during the early stage, but decreased during the late stage [101]. While normal vasoregulatory responses help resolve low levels of inflammation, chronic inflammation can even impair vasoregulation, which causes the changes in the blood flow observed during the progression of IBD [102]. Abnormal lymph flow through the lymphatic vessels was also observed during IBD along with reduced lymphatic vessel contractile activity [103]. This impaired lymph flow can promote accumulation of inflammatory components in the interstitial space, perpetuating the inflammatory environment. Correspondingly, VEGF-C induction that enhances the lymphatic density and function has been shown to have therapeutic effects against IBD by clearing inflammatory cells and bacterial antigens and by modulating the macrophage activities [104]. Furthermore, increased adhesion of the circulating cells on the endothelium is commonly found in a variety of IBD models corresponding to the increased expression of different endothelial cell adhesion molecules [105,106,107]. This corresponds to the altered leukocyte trafficking observed in IBD patients [108].

The impaired intestinal vascular function is also related to the pathogenesis of colorectal cancer. The lymphatic endothelium is affected by colorectal tumors to promote the tumor metastasis [109]. In contrast to the protective function of VEGF-C in IBD models, the same growth factor impaired the lymphatic vessel barrier and facilitated the entry and metastasis of colorectal tumors [110]. Given the close relationships between these diseases and gut microbiome dysbiosis, it also would be interesting to study the interactions between the gut microbiome and the intestinal vasculatures. Indeed, previous studies reported the roles of gut microbiome on intestinal vascular development and integrity [111,112,113,114]. Further investigation into how the microbial antigens, metabolites, and invading gut microbes affect the lacteals in homeostatic and pathological states would give new insights into the progression of diseases and new potential targets to resolve the pathological conditions.

### 4.3. In Vitro Models of the Intestine

#### 4.3.1. Current In Vitro Models of the Intestine

Despite ardent research efforts in the intestine due its diverse functions in the human body [115,116,117,118], we still lack the mechanistic understanding of many aspects of the intestinal environment. While animal models enable systemic analyses of the intestinal environment, it is hard to investigate the contribution of each component in the intestine to the observed responses. The cause and effect relationships between the components also remain unclear. Furthermore, the results from animal models are not necessarily translated into humans and inter-individual variations were not considered. Thus, modeling the intestine in vitro is critical to advance our understanding of its complex environment and the pathogenesis of various gastrointestinal and systemic diseases affected by the intestinal functions. The rapidly growing interest in the gut microbiome further necessitates appropriate experimental systems and tools to understand more deeply host–microbiome interactions.

Indeed, many 2D and 3D in vitro models have been developed to recapitulate the physiologically similar environment of the intestine. Various designs using Transwell inserts, organoids, and biomaterial scaffolds have been explored with different cell types, biomaterials, and scaffold shapes. These advanced models successfully recapitulated the heterogeneous cell population of the intestinal epithelium, as well as the villi-crypt architecture. While more detailed review of the previous in vitro models of the intestinal epithelium is beyond the scope of this review and can be found elsewhere [119,120], these static models do not fully recapitulate the dynamic environment of the intestine. As a platform for the recapitulation of the dynamic, complex environment of organs, microfluidic devices have gained increasing attention. Microfluidic chips, referred to as “organs-on-chips,” have been applied to mimic many organs, including lung, heart, and the BBB (discussed further later in this review). These devices enable mechanical flows that can exert important influences on the cell phenotypes. Different media can be flown through separate channels to co-culture multiple types of cells and create tissue–tissue interfaces [121], and attempts to connect the chips of different organs have been reported to study the organ–organ interactions [122,123].

For the intestine, microfluidic chips were used to further mimic the biomechanical environment of the intestine that can affect the growth, differentiation, and function of the epithelial cells. Microfluidic chips with two channels separated top and bottom enable independent control of the flow and the fluid composition [124]. Furthermore, the cyclic mechanical deformations that mimic the peristalsis of the intestine were created by the cyclic suction in the hollow chambers attached to the sides of the channel (see Figure 5d). In this design, flow control could be decoupled from the control of the mechanical deformations, allowing investigation of the independent effects of the intestinal fluid flow and the peristalsis movement on the bacterial growth. The chip also recreated the biomechanical environment observed in the native intestine. Caco-2 cells, which are commonly used human intestinal epithelial cell lines, have been successfully differentiated into absorptive, goblet, enteroendocrine, and Paneth cells and formed the villi structure of the small intestine [125]. More advanced microfluidic chips have been developed to incorporate other features of the intestine. Lipopolysaccharides (LPS) endotoxin and human peripheral blood mononuclear cells (PBMCs) were added in the chip to model intestinal inflammation [126]. The complex community of both aerobic and anaerobic human gut microbiome could also be co-cultured with the human intestinal cells due to the oxygen gradient established by the flow of oxygenated medium in the lower channel inside an anaerobic chamber [127]. A different study further added a pathogenic bacteria, *Shigella*, to a gut-on-a-chip demonstrating that flow and the cyclic mechanical strain can affect the infection activity of the bacteria [115]. While most of the gut on-chip models used Caco-2 cells, which do not transcriptionally mimic the primary intestinal epithelial cells, the combination of the organoids and microfluidic chips was suggested to benefit from the both in vitro models. The enteroid fragments seeded in the chip successfully differentiated and formed the villi structure whose cells were more transcriptionally similar to those in vivo than the organoids alone [116]. This suggested a possible application of gut-on-chips in personalized medicine. Furthermore, the mechanism of how the mechanical forces affect the villi morphogenesis was investigated on gut-on-chips. The basal flow that constantly removes a Wnt-antagonist and induces the expression of the Wnt receptor has been revealed to be a major mediator of the intestinal villi morphogenesis [117]. These studies showed great promise for intestine-on-chips in various applications, such as personalized medicine, drug testing, and the studies of intestinal morphogenesis and human-gut microbiome interactions.

#### 4.3.2. Intestinal-Vasculature-On-A-Chip

Despite the continuous development of intestine-on-a-chip models that faithfully mimic the structure and functions of the intestinal epithelium, the physiologically relevant recapitulation of the intestinal vasculature has yet to been achieved. The previously developed gut-on-chips incorporated the intestinal endothelium in the lower channel and revealed that the endothelium is important for the maintenance of the intestinal barrier and the production of the mucus [116,127]. Especially in disease conditions, the contents in the plasma transported by the intestinal blood vessels could affect intestinal epithelium functions [118]. However, these previous models failed to mimic the physiological structures of intestinal microcirculation. The microvascular ECs were not tested for their identity as either vascular or lymphatic cells. Furthermore, the distance between the epithelium and the endothelium differed from that in vivo. Given the importance of the intestinal vasculature on intestinal function, interactions with the human gut microbiome, and its implications in various gastrointestinal diseases, the intestine-on-a-chip model should also successfully mimic both the structures and the functions of the intestinal vasculature and have them validated. Although no study has yet focused on modeling the intestinal vasculature, some previous studies suggest useful information for the intestinal-vascular-on-a-chip designs.

While the typical microfluidic chips were made by soft lithography using polydimethylsiloxane (PDMS), another fabrication method of an organ-on-a-chip was suggested to enable the incorporation of multiple intestinal layers. The chip was manufactured following a “cut and assemble” method using thermoplastic polymethyl methacrylate (PMMA) (see Figure 5a). Each layer was processed using CAD software, created with laser cutting technology, assembled via adhesive layers, and seeded with a different type of cell [128]. While the authors did not attempt to seed vascular endothelial cells, this modular assembly method that enables an independent design of the channels in each layer shows promise in creating a physiologically relevant epithelium–endothelium interface in the intestine

The matrix for the support of the cell adhesion and migration inside the channel both need to be further modified to mimic the interstitial space. Collagen and Matrigel, or the combination of both is often used to coat the porous membrane onto which the cells are seeded in the chips. Additionally, the membrane that separates the intestinal epithelium from the endothelium is 20 µm thick, which is different from the physiological distance of the villi capillaries (2 µm) and lacteals (50 µm) from the epithelium [70]. Such differences can affect the analyses of drug absorption and transport from the intestine. Furthermore, the vasculature formation and transcriptome of the ECs are affected by the biochemical and biomechanical properties of the extracellular matrix (ECM) [129]. The composition, thickness, and stiffness of the ECM should be adjusted to the endothelial cells to recapitulate each cell’s behavior observed in the native tissue. The design of the intestinal vasculature-on-a-chip should consider organ-specific and cell type-specific matrix materials.

More biomimetic vasculature models can be created by seeding the ECs around the inner surface of the channel instead of the previous monolayers seeded on the top of the porous membrane in the chip used for 2D analyses. This will create a 3D cylindrical system which is more physiologically relevant to the tubular structures of the blood and lymphatic vessels (see Figure 5b). A 3D organotypic model of tubular blood vessels was previously created to study the interactions with the pancreatic cancer cells and the blood vessels [130]. The same method can be further applied to create biomimetic lymphatic vessels.

Interestingly, a microcirculation-on-a-chip to incorporate both blood and lymphatic vessels has been reported (see Figure 5c). The flow facilitated the formation of intact vessels and the structures and functions of the generated vessels were validated [131]. This suggests that blood and lymphatic ECs can be co-cultured in a microfluidic device to form vessels. A physiologically relevant intestine-on-a-chip with the epithelium and both vasculatures would provide new insights into how these compartments interact in the intestine during homeostasis and diseases. Although yet to be achieved, its development could improve the studies of colorectal cancer metastasis, the systemic effects of the gut-derived nutritional and microbial signals, and the delivery methods of drugs and emerging immuno-engineered particles. This model would also enable the discovery of complementary therapeutics that target these vessels to resolve disease conditions that prevent the reduction of efficacy of existing therapeutics by the abnormal microenvironment during diseases.

## 5. Brain Microenvironment and In Vitro Models

For both the cardiovascular and lymphatic circulatory systems, organ-specific anatomy and physiology is a common feature. As mentioned above, variations in endothelium structure, mural cell coverage, and receptor expression can be found throughout the body to allow the circulatory systems to best serve the organ at hand. In the case of the brain, a metabolically active, pressure sensitive, and highly protected organ, vessels must be blocking harmful biochemical and physical agents while simultaneously providing nutrients and clearing waste to support homeostasis without impeding neural function. This calls for close cooperation of the cells involved in neural function, i.e., neurons and astrocytes, with those involved in circulation (ECs and mural cells).

### 5.1. Structure and Functions of the Brain

The central nervous system (CNS) is composed of the brain and the spinal cord, which can be viewed as the main processor and controller of the body, respectively. Through close coordination with the peripheral nervous system, the brain processes an overwhelming majority of external and internal stimuli and enacts changes to the body through both electric neural activity and neuroendocrine signaling. This is accomplished through the constant firing of action potentials by neurons, an energetically expensive process that involves the transport of ions and the release of neurotransmitters, such as glutamate, norepinephrine, and nitrous oxide (NO), into synapses. Synapses are the extracellular space between neurons that allow for signal transduction in neural networks. Astrocytes work alongside neurons as a type of glial cell that actively supports neural activity largely through formation, maintenance, and elimination of synapses. Astrocytes are responsible for regulating ions and clearing neurotransmitters from the synaptic space and can receive signals directly from the neurons with which they are closely associated.

### 5.2. Role of Blood Vessels in the Brain

With its constant activity and critical role in human life, the brain has specific circulatory mechanisms to ensure that it continuously receives proper nutrients without negatively impacting neural activity. A network of pial arteries and veins covers the surface of the brain. The pial arteries branch off from the network and journey deep into the brain while the surfacing intracortical veins join the network, creating a honeycomb-like structure [135]. The intracortical arteries further branch into arterioles and numerous capillaries, creating a dense microvascular network within the brain. This capillary network is tightly co-localized with most neurons, with a microvessel being located no further than 15 µm from the soma of every neuron (excluding perivascular areas around large vessels, as it is believed that oxygen and glucose can passively diffuse directly to the surrounding tissue in such areas). As mentioned earlier, these capillaries have a continuous endothelium that allows strict regulation of metabolites, nutrients, and other molecules in the brain microvasculature and creates the blood–brain barrier (BBB). The BBB’s main role is preserving and supporting homeostasis in the brain, but it has other important functional aspects, such as preventing toxins, pathogens, and other potentially harmful substances from reaching the brain and the alleviating injury, inflammation, and disease. Beyond simply having a complete, non-fenestrated endothelium, these capillaries have numerous other properties such as the leukocyte adhesion protection, specialized tight junctions, and inhibited bulk-flow diffusion and transcytosis including pinocytosis [1,124].

Many of the specialized properties of the microvasculature are maintained through close interactions with and signaling from pericytes [136]. The BBB endothelium, pericytes, neurons, and the endfeet of astrocytes comprise the neurovascular unit (NVU) (see Figure 6). Within the NVU, neurons, pericytes, and astrocytes all play important signaling roles in regulating blood flow. In general, more active areas of the brain require greater blood flow to supply the metabolically active neural tissues with oxygen, glucose, and other molecules necessary for neural activity [137]. Neurons generate signals via glutamate activity to initiate both direct interactions and interactions with other cells such as astrocytes. For direct interactions, glutamatergic synaptic activity increases intracellular Ca^2+^ concentration and activates Ca^2+^-dependent enzymes, such as cyclooxygenase 2 (COX-2) and neuronal NO synthase (nNOS), leading to the production of two potent vasodilators, prostanoid and NO, respectively. Furthermore, the glutamate produced for the direct neural signaling also acts on the metabotropic glutamate receptors in astrocytes, creating another cascade of intracellular Ca^2+^ increase and vasodilator production, namely the prostaglandin E_2_ (PGE_2_) and epoxyeicosatrienoic acids (EETs). PGE_2_ also directly affects pericytes through their E-type prostanoid receptor 4 (EP4) receptors and causes pericytic relaxation [124]. In contrast, pericytic contraction has been shown to be affected by norepinephrine. Since the processes of pericytes directly cover between 30% to 90% of the capillary walls in microvessels, any relaxation or contraction of pericytes can lead directly to an increase or decrease in capillary diameter, respectively. Pericytes have also been shown to have a role in angiogenesis through interactions between platelet derived growth factor B (PDGFB) and its receptor (PDGFRβ), as well as through transforming growth factor-β (TGFβ) and its receptor (TGFRβ2), sphingosine-1 phosphate (S1P). PDGFB secreted by the endothelium is essential for recruiting undifferentiated perictyes to cover newly formed vessels [30]. Together with Notch signaling, the TGFβ/TGFRβ2 interaction promotes pericyte differentiation, the attachment of pericytes to the new vessels, the formation of the shared basement membrane between endothelial cells and pericytes, and the inhibition of further endothelial cell proliferation in order to stabilize the new vessels. Specific to microvasculature in the brain, pericytes also secrete angiopoietin (Ang-1), which is received by its endothelial receptor Tie2 to promote the formation of the BBB. With a similar effect, S1P’s receptor on pericytes downregulates vascular permeability genes and promotes the formation of both pericyte-endothelial (N-cadherin) and endothelial-endothelial (VE-cadherin) interconnections [124].

These cell interconnections are integral to the proper function of the BBB. Accordingly, ECs in the brain have highly complex and specialized tight junctions, similar to the tight junctions of epithelial cells, that provide the basic integrity of the BBB. However, it is believed that these tight junctions would not be able to exist without adherens junctions and the crosstalk between these two junction types. Tight junctions and adherens junctions each have distinct molecular components and separate roles. Adherens junctions consist of cadherins, largely VE-cadherins in the brain, and form before tight junctions and initiate cell–cell interactions, which regulate the maturation and some physical properties of the cell [138]. Tight junctions are formed from occludin, a multiple claudins (claudin-3, claudin-5, and claudin-12), and members of the junctional adhesion family, including JAM-A, JAM-B, JAM-C, and JAM-4 [138,139]. These transmembrane proteins connect to the cell’s actin cytoskeleton via Zonula occludens-1 (ZO-1). Once formed, they have two main functions that create the highly selective and protective BBB, the “gate” function and the “fence” function. On the one hand, the “gate” function is the ability of the tight junction to regulate use of the paracellular route, and thereby allowing high selectivity of which solutes and ions are diffused. On the other hand, the “fence” function establishes cell polarity by limiting which lipids and proteins are able to move freely from both the apical and basolateral surfaces of the cell. While most tight junctions in the BBB exist between only two ECs, molecularly-distinct tight junctions are found at tricellular connections. These specialized junctions have tricellulin and lipolysis-stimulated lipoprotein receptor to potentially regulate paracellular permeability [139].

### 5.3. Role of Lymphatic Vessels in the Brain

There are four major fluid compartments in the brain which include: intracellular fluid, the blood vasculature (see above), cerebrospinal fluid (CSF), and interstitial fluid (ISF). On the one hand, the CSF is generated by the choroid plexus and acts as both a protective fluid layer in the brain and the supply stream for various molecules needed to maintain homeostasis, such as neuroendocrine signals. On the other hand, the ISF is largely a collection of metabolic wastes and other solutes that need to be cleared from the brain. The CSF enters the brain and moves into the parenchymal space, which in turn causes the ISF to move into the lymphatic system of the brain. However, the brain does not have a lymphatic system in terms of the classical system defined earlier. Rather, it has perivascular pathways for both interstitial fluid (ISF) and cerebrospinal fluid (CSF) bulk transport, called the glymphatic system. The glymphatic system is paired with the meningeal lymphatics system, a pathway that allows the drainage of ISF and CSF from the CNS to the peripheral vascular system and lymph nodes. Unlike the meningeal lymphatics system, the glymphatic system lacks LECs entirely. Instead, the CSF and ISF flow through perivascular spaces bound by large vessels, i.e., arteries/arterioles for CSF and veins/venules for ISF, on one side and the endfeet of astrocytes on the other [140]. Perivascular aquaporin-4 (AQP4) appears to play a large role in the maintenance of this pathway, and, in healthy conditions, displays very specific, polarized expression in astrocytic endfeet. The importance of AQP4’s role in the glymphatic system has been supported by in vivo studies, in which deletion of AQP4 has been shown to reduce the clearance of extracellular molecules from the perivascular space in the brain and lessen the CSF-ISF exchange [140].

The structure of the glymphatics and meningeal lymphatics system varies throughout the brain. For instance, the perivascular spaces of the glymphatic system varies with blood vessel diameter. For larger vessels, there exists a particular channel called the Virchow–Robin space, which is lined with leptomeningeal cells on both the inner vessel border and the outer astrocytic endfeet border. The Virchow–Robin space narrows as it follows the vessels and eventually ceases to exist at the capillary level. Rather than using a dedicated space, the fluid flows through the extremely porous basal lamina as part of the NVU at the capillary level, where the space is still bounded by the endothelium on the interior and the astrocytic endfeet on the exterior. Composed mostly of laminin, type IV collagen, and fibronectin, the basal lamina is a thin layer of extracellular matrix that tightly links all cell types in the NVU through integrins and proteoglycans, amongst other adhesion molecules [141]. On the contrary, the meningeal lymphatics system has dedicated lymphatic vessels that are found mostly in the dura mater, the outermost layer of the brain that is adherent to the skull. These lymphatic vessels display regular capillary markers, including LYVE1 and CCL21, and drain the brain of macromolecules as well as immune cells. Similar to classically defined lymphatic vessels, it then drains the immune cells to the cervical lymph nodes and clears wastes to the circulatory system to be processed and excreted. The inferior portion of the dura mater appears to have a much more extensive network of lymphatic vessels than the superior portion, though vessels in both portions contained semilunar valves despite the fact that lymph flow at the base of the brain is largely directed by gravity, and therefore should not require valves to prevent backflow [140].

### 5.4. Diseases of Brain Vasculature

While many diseases have vascular implications or effects, some of the better understood and more popular diseases include stroke, multiple sclerosis, trauma, and a variety of neurodegenerative diseases, including Alzheimer’s disease (AD).

AD is the most common cause of dementia and a devastating problem for both those afflicted with the disease and their loved ones. Through extensive imaging techniques, the physiological environment of AD is becoming better understood, but much about the disease still remains unknown. It is known however, that AD is characterized by the formation of plaques that consist of amyloid-β (Aβ) oligomers and tangles that are comprised of tau protein. Monomeric Aβ is regularly cleared from healthy individuals, but a reduction of clearance and a subsequent increase in aggregation lead to oligomerization and eventual polymerization in AD patients, causing plaque formation [132]. The specific structure of these plaques is amyloid fibrils with tight β-pleated sheets. Similarly, tau typically exists as an unfolded random coil protein associated with neuronal microtubules, but it also folds into fibrils with β-pleated sheets in the neuronal cytoplasm of AD patients. Both structures interfere with proper neural function, leading to worsening memory loss, and neuronal loss. These toxic plaques and tangles often spread throughout the brain, further impairing cognitive function [142].

Little is understood as to why the Aβ oligomers form and are not cleared from the brain. Currently, the most popular theory is that the oligomers are caused by less efficient processing of amyloid precursor protein (APP). Normally, the enzymatic cleavage of APP by γ secretase form Aβ peptides. When Aβ monomeric peptides form in healthy individuals, phagosomes, lysosomes, and ubiquitin-protease pathways break them down and allow them to be cleared from the brain [133]. Consequently, the CSF can contain higher levels of Aβ in AD patients, although the amount is still so low that it is difficult to detect. Clearance issues can be in part due to problems with AQP4 expression, as it has been seen that Aβ can cause irregular AQP4 expression in astrocytic endfeet [134]. Without AQP4, fluid pressure is not maintained in the glymphatic system, and thus, Ca^2+^ concentrations rise to neurotoxic levels and neuroinflammation occurs [143]. These issues can further be associated to nonfunctional glutamate reuptake, which is largely controlled by AQP4 and makes AQP4 an attractive target for AD therapies [144]. In addition to AQP4, low-density lipoprotein receptor-related protein 1 (LRP1) and P-glycoprotein (P-gp) deletions can impair the clearance of Aβ. While LRP1 deletion studies in mouse models have reorted mixed results both supporting and refuting LRP1’s involvement in Aβ clearance, the role of P-gp is more widely supported. Animal studies have shown that P-gp deletion impairs Aβ clearance and leads to Aβ deposition in the brain, as well the fact that P-gp expression is reduced near amyloid plaques. Clinical data also support the role of P-gp in AD pathology, as positron-emission tomography (PET) studies have demonstrated that the P-gp transporter receptor is compromised in the BBB of AD patients and epidemiologic studies have illustrated an inverse correlation in Aβ plaque numbers and P-gp expression in patients without dementia [145].

Beyond complications caused by the Aβ plaques and tau tangles in AD, other more clearly vascular issues are present in AD patients. Magnetic resonance imaging (MRI) studies have shown some increase in BBB permeability, but other imaging modalities have failed to confirm an increase in BBB permeability. Other MRI studies have also illustrated a global, severe decrease in cerebral blood flow in AD, which is often preceded by reduced cerebral blood flow in specific portions of the brain, i.e., the posterior cingulate gyrus and precuneus, frontal and occipital cortices, parahippocampal gyrus, hippocampus, and entorhinal cortex, in early stage AD or age-related mild cognitive impairment [146].

### 5.5. In Vitro Models of Brain Vasculature

In vitro models for both healthy and diseased neurovasculature have increased in popularity in recent years. Some of the advantages of in vitro models include the use of human cells, reduced use of animal models, and the high level of environmental control that offer physiological relevance, ethical treatment of animals, and investigative specificity advantages, respectively, as well as a general monetary advantage by reducing the cost of expensive animal trials in the earlier stages of investigation. As mentioned earlier, in vitro models can be 2D or 3D, with physiological relevance more closely seen in 3D models than 2D models. However, this does not mean that 2D models are not useful. For example, immortalized human brain microvascular endothelial cells have been grown in both 2D single culture and 2D co-culture with primary astrocytes and pericytes to model the BBB functions. This model allows the cells to be plated with and without stressors and explore possible drug targets. Under healthy conditions, the 2D co-culture formed complete meshwork-like networks seen in healthy tissue in vivo. After illustrating that the co-culture model displayed relevant angiogenic behavior, oligomeric amyloid-β (oAβ), a common stressor in AD, was added to the model in one of the following two ways: (1) Simultaneous plating of the cells and the oAβ (disruption of meshwork formation paradigm) and (2) plating the oAβ after the cells formed their meshwork (disruption of performed meshwork paradigm). In both paradigms, the total cell coverage of in the networks was 16% to 20% less and the number of meshes was decreased by 40%, indicating reduced angiogenesis. To investigate targets to prevent the meshwork disruption following oAβ, a known pro-angiogenic factor, epidermal growth factor (EGF), was added to the cells during plating as a preventative treatment before the stressor was added following some meshwork formation. EGF prevented the meshwork disruption seen in previous trials, therefore, EGF was, then, included in an AD mouse in vivo model where the EGF treatment was shown to reduce the AD-induced angiogenic changes, such as meshwork disruption [147].

While 2D in vitro models offer simple ways to explore disease physiology and cause, 3D models have the advantage of more accurately recreating the microenvironment. There are many types of 3D models for neurovasculature, ranging from 3D spheroids to microfluidic chip models. Three-dimensional spheroids have the added advantage of transitioning from in vitro models to in vivo models with implantation. One example includes a technique used to model the BBB, in which immortalized hCMEC/D3 (human cerebral microvascular EC line D3) and primary human brain microvascular ECs (HBMECs) were both co-cultured with primary human astrocytes and human brain vascular pericytes (HBVP) to recreate NVUs without the neurons. Ninty percent of these spheroids self-assembled into acceptable spheroids within 48 h, and they displayed tight junctions, which are critical to the BBB’s primary barrier function. Subsequently, these spheroids were analyzed for various receptor expression and, then, used for early-stage drug testing to discern the ability of the drugs of various size to pass through the BBB in normal physiological conditions [148].

While the previous model is certainly a useful tool in neurovasculature studies, it must be noted that analysis of the ultrastructure of the spheroids revealed an inverse layering of the ECs, pericytes, and astrocytes. Whereas NVUs in vivo have the ECs surrounded by pericytes and then loosely covered by astrocytes, these spheroids had astrocytes clustered mostly in the center with ECs as the outermost layer and pericytes as the layer in between, attached to the ECs. To control for cell patterning during layer formation, microfluidic on-chip models are an attractive choice. One group used a 3D microfluidic chip model with two separate channels to model the BBB and recreate the anatomy seen in vivo. They used both co-culture models, ECs and pericytes, as well as triculture, ECs with pericytes and astrocytes, to successfully recreate the barrier function seen in the brain by discriminating various makers based on their size. In addition, the triculture model even displayed tight junction pores with similar diameter size to those found in vivo, demonstrating a physiologically accurate recreation of the BBB pore size. In addition, the triculture model also displayed an increase in functional expression of the P-gp efflux pump over time while in culture, further solidifying its identity as a robust model of the BBB [149].

Some chips have been able to recreate the distinctive AQP4 polarized distribution pattern seen in NVUs, which is an important feature for the integrity of the BBB. One group used a chip with an upper and a lower layer separated by a porous membrane with immortalized human brain microvascular endothelial cells, human brain vascular pericytes, and human astrocytes. This design allowed blood to continuous flow over the top of the endothelial monolayer on top of the membrane. The underside of the membrane had a chamber of pericytes lining the membrane and astrocytes suspended in Matrigel, with their AQP4-expressing endfeet positioned ideally in the 3D space to interact with the pericytes directly while still being near the blood vessel. This unique porous membrane between the chambers facilitated the paracrine and juxtacrine signaling between all three cell types. In addition, this model enabled 3D mapping of nanoparticle distributions in both the vascular and perivascular chambers, which is important for evaluating the cellular uptakes and penetrations of the BBB through distinct receptor-mediated transcytosis pathways [150].

More complex chip devices have also been created that reproduce the Aβ plaques seen in AD patients, as well as both the tight and adherens junctions. For example, a five-channel microfluidic device has been created. On one half of the device is the ReNcell chamber that consists of a microchannel of ReNcell VM human NPCs (an immortalized cell line of neuroprogenitor cells) in Matrigel and an adjacent microchannel containing ReNcell differentiation media. The other half of the device has a BBB chamber that consists of a microchannel lined with BECs containing BEC media and a collagen scaffold microchannel. After seeding and establishing the two chambers separating, the intervening barrier microchannel is filled with collagen, which allows the two chambers to interact. For AD pathophysiology, ReNcells with familial AD (FAD) mutations of genes responsible for Aβ plaques were seeded. Chips with the AD cells had reduced BBB permeability, as seen in AD patients, as well as other AD-specific features, such as decreased expression of both claudin-1 and claudin-5, decreased VE-cadherin expression, increased expression of matrix-metalloproteinase-2, increased expression of reactive oxygen species, and the deposition of Aβ peptides at the endothelium [151].

## 6. Conclusions

In vitro microphysiological systems provide integral platforms to investigate complex physiological mechanisms that cannot be easily studied in in vivo models. Each of the 2D and 3D systems has their own advantages as shown in Table 1 depending on a variety of applications. In this review, we placed a special focus on the vascular system and the use of the microfluidic system to model the vasculatures in vitro. On-chip platforms for vasculatures have proven especially valuable to recapitulate the conduit structure of vessels, in a dynamic microenvironment by incorporating and allowing the independent control of the physiological flow (luminal and interstitial). The on-chip platforms would enable researchers to investigate structural remodeling of tissues in various conditions as well as tissue–tissue and cell–cell interactions in vitro, often with human-derived tissues and cells. However, limitations in the current chip models still remain. The common chip fabrication method does not always result in chips with the highest quality, mostly in terms of difficulty with culturing multiple cell types in the ”universal” media, obtaining primary vascular endothelial cells and mural cells from specific organs, and seeding cells in controlled manners in complicated 3D structures. Moreover, the protocols for creating chips with multilayered cells are yet to be established and standardized to better recapitulate multilayered vasculatures. Although many chip models have been developed for various organs, there do not exist as many chip models for vasculatures in specific organs despite the endothelial cell heterogeneity. We expect that a more complete entire body-on-a-chip system could be enabled by the development of organ-specific vasculatures chips.

## Figures and Tables

**Figure 1 micromachines-11-00147-f001:**
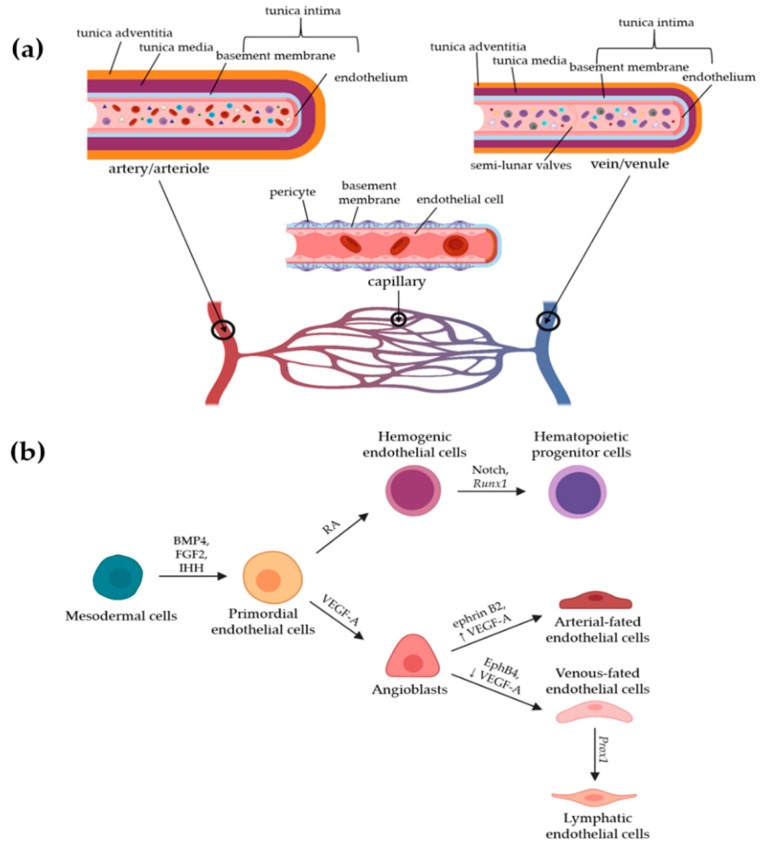
(**a**) The distinct structures and general functions of the three main types of blood vessels: arteries/arterioles, capillaries, and veins/venules; (**b**) The differentiation pathway of the major cell types in the cardiovascular system as well as lymphatic endothelial cells (ECs). The mesoderm is the source of the precursor vessel cells and hematopoietic progenitor cells. From there, precursor vessel cells differentiate into either arterial-fated or venous-fated ECs. In the lymphatic vasculature, venous-fated ECs further differentiate into lymphatic ECs (LECs).

**Figure 2 micromachines-11-00147-f002:**
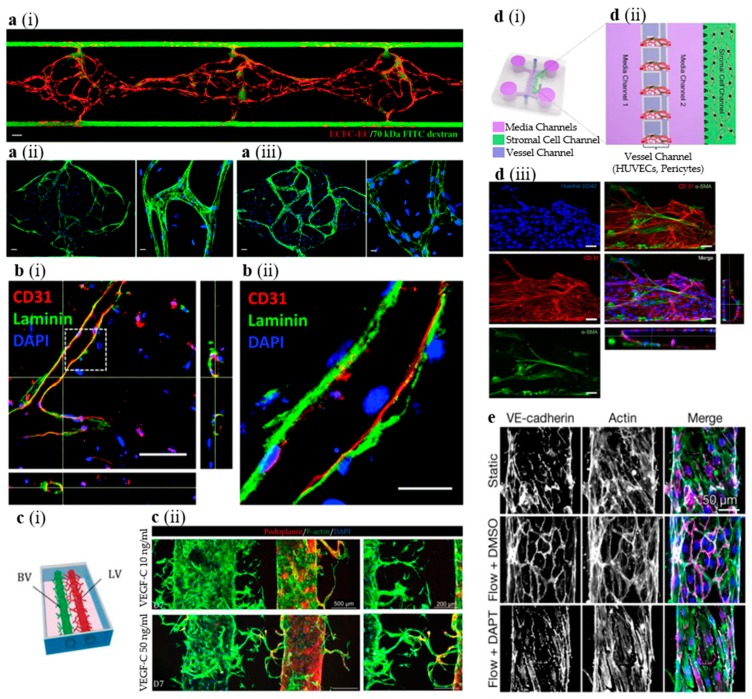
A collection of blood vasculature on-chip in vitro models that recreate different aspects of blood vessel anatomy and physiology. (**a**) The multiwell tissue flow chambers of Hughes lab’s on-chip model illustrate (i) the EC (mCherry) vascular networks formed after 7 days and the flow of 70 kDa FITC-dextran 30 min after perfusion, (ii) the distribution of Claudin-5 (Alexa Flour 488) and nuclei (4× and 20× magnification), and (iii) the expression of VE-cadherin junctions (Alexa Flour 488) and nuclei (DAPI) (4× and 20× magnification). Scale bar = 50 µm [17]. (**b**) The iPSC-EC microvessels display complete lumens with laminin deposition in the basement membrane, Scale bar = 100 µm, in (i) and 25 µm in (ii) [18]. (**c**) A double vessel chip (i) with separate channels for blood vessel (BV) and lymphatic vessel (LV) shows the effect of different VEGF-C concentrations on angiogenesis and lymphangiogenesis [19]. (**d**) A multichannel chip (i) overview schematic and (ii) cross-sectional schematic illustrating the layering of the pericytes (stromal cells) with the endothelial cells and fluid media channels. (iii) Immunostaining with confocal microscopy reveals the architecture of the microvessel including complete lumen formation and pericyte incorporation through Hoechst 33,342 staining of the nuclei (blue), CD31 as an EC marker to illustrate angiogenesis (red), and α-SMA (smooth muscle actin alpha) as a marker of pericytes (green) following fixation 8 days after seeding. Scale bars = 40 µm [20]. (**e**) Immunostaining for VE-cadherin (magenta), labelled with DAPI (blue), and the actin stain phalloidin (green) as compared with the structure of human engineering microvessels (hEMVs) in static conditions and exposed to flow containing either dimethyl sulfoxide (DMSO) or DAPT (a Notch signaling inhibitor). Scale bar = 50 µm [21]. Figure republished with permission from each indicated reference as follows: [17] for part (a), [18] for part (b), [19] for part (c), [20] for part (d), and [21] for part (e).

**Figure 3 micromachines-11-00147-f003:**
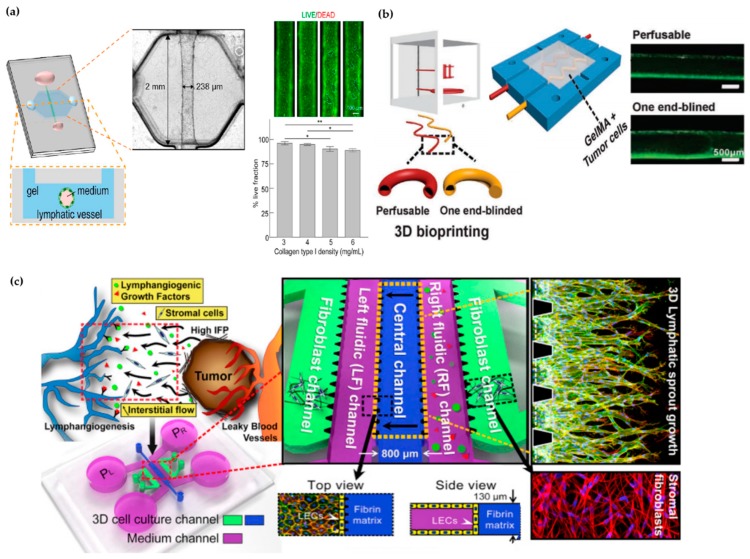
Lymphatics-on-a-chip: (**a**) A single lymphatic vessel in a microfluidic system [55], (**b**) bioprinted perfusable blood vessel and one-end-blinded lymphatic vessel on a microfluidic chip [56], (**c**) Generation of interstitial flow using pressure gradient created by the volume difference between the two fluidic channels around the central channel [57]. Figure republished with permission from each indicated reference as follows: [55] for part (a), [56] for part (b), and [57] for part (c).

**Figure 4 micromachines-11-00147-f004:**
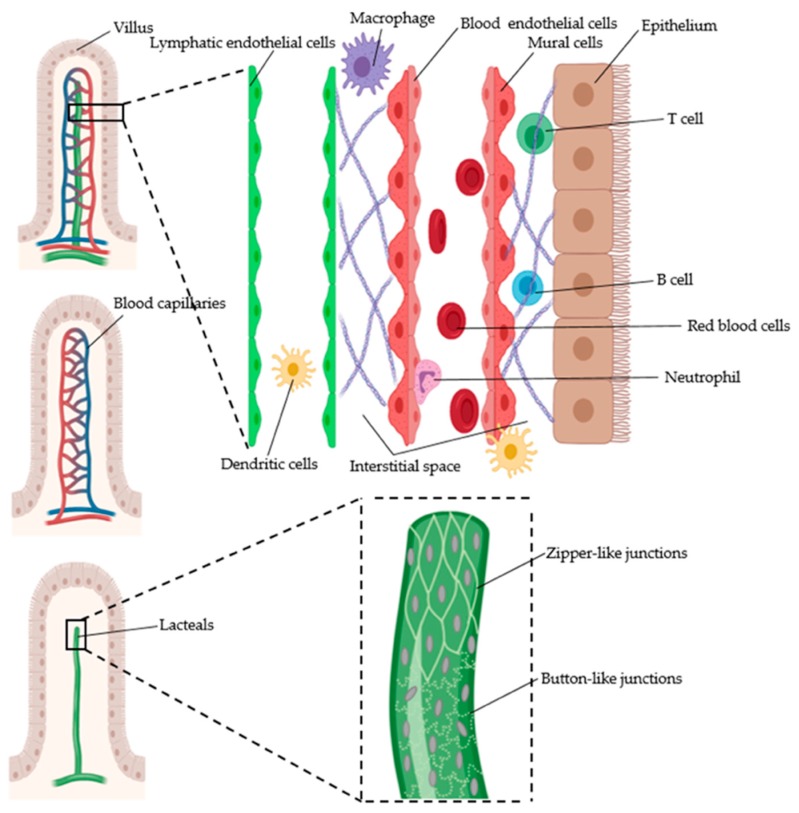
Vasculatures in a villus of the small intestine.

**Figure 5 micromachines-11-00147-f005:**
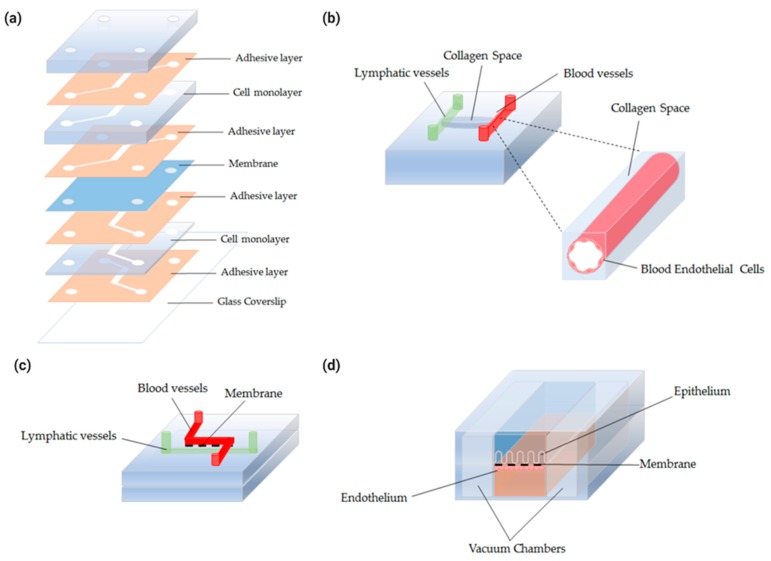
Microfluidic chip designs for a potential intestinal-vasculature-on-a-chip: (**a**) The “cut and assemble” method can be used to independently design the channel of each layer [132], (**b**) the 3D cylindrical system can be used for more physiologically relevant model of the intestinal vasculatures for 3D analyses [133], (**c**) blood and lymphatic vessels can be formed on a single chip [134], and (**d**) vacuum chambers generate cyclic mechanical strain [124].

**Figure 6 micromachines-11-00147-f006:**
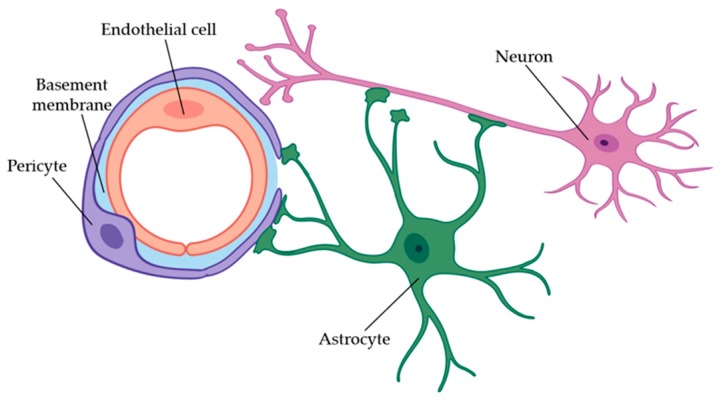
A simplified cross-sectional diagram showing the structure of a neurovascular unit in the brain and illustrating the cell–cell interactions. The innermost layer consists of the mircovessel endothelial cells surrounded by the basement membrane. Pericytes attach to and incompletely cover the basement membrane. The endfeet of astrocytes act as the outermost layer of the vascular structure and interact directly with the other cells of the neurovascular unit (NVU). Neurons can interact with the other cells either directly through their dendrites and subsequent synaptic space or through the astrocytes.

**Table 1 micromachines-11-00147-t001:** Two-dimensional (2D) vs. three-dimensional (3D) in vitro models.

Gut		Brain	
2D	3D	2D	3D
Transwell Inserts [116,117,128] Cell monolayer on top of ECM-coated porous membrane Static medium on top of the monolayer Dynamic flow of medium in the bottom of the monolayer is possible with a flow generator Villus-crypt morphology enabled with flow Commonly used for drug permeability tests High reproducibility and ease of use Short lifespan	ECM Scaffolds [152] Cell culture completely embedded in ECM or bio-fabricated in desired shapes such as villus-crypt or tube Simple 3D models for Cell-ECM interactions Can easily recapitulate physiologically relevant architecture Organoids [116,119,120] Intestinal stem cells from crypts or induced pluripotent stem cells cultured in 3D ECM gels Villus-crypt morphology Differentiation of stem cells into different types of the intestinal epithelial cells mimicking the epithelium cell composition Cannot access the luminal side independently from the outside of the epithelium Short lifespan Microfluidic Chips [117,120,123,125,127] Independent control of the inputs and outputs to the luminal side of the intestine Spatiotemporal control of both the biochemical and biomechanical microenvironments Villus-crypt morphology enabled with dynamic microenvironment Recapitulation of both peristaltic motion of the intestine and the luminal flow Incorporation of other biological components (gut microbes, immune cells, other tissues)	Endothelial Monolayers [147,149] Single layer of either immortalized or primary ECs Displays both tight and adherens junctions, important for BBB specificity Either static or flowing media conditions Integral for perfusion, drug diffusion, and other permeability assays Co-culture with pericytes allows direct cell-to-cell and tissue interactions. Simple control of stressors and other environmental factors	ECM Scaffolds [150] Allows recreation of local 3D microstructure Can work with other 2D or 3D applications, such as monolayers or microchannels Simple and able to be cast or fabricated into shapes Organoids [14,148] Allows 3D layering, including perivascular and vascular spaces Easily converted to in vivo model through implantation Self-assembly into inverse NVU structures Can be integrated into flow devices Microfluidic Chips [150,151] Accurate biological 3D structure Continuous physiological flow with precise control of inputs and outputs, including pathological and drug components Endfeet-specific expression of AQP4 for specific 3D microstructure Short lifespan Recapitulation of amyloid-β plaque deposition as seen in AD patients

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
