# Peer review of "Blood and Lymphatic Vasculatures On-Chip Platforms and Their Applications for Organ-Specific In Vitro Modeling"

_micromachines, 2020, doi:10.3390/mi11020147_

Round 1

Reviewer 1 Report

The manuscript reviewed lymphatic and blood vasculature systems in a point of physiology and in-vitro model. It  will be taken attention to the audience of the micromachines. However, following issues the reviewer raised should be discussed before publication.

 [1] The manuscript focused on physiology (or biology) explanation instead of in-vitro model. Tittle did not present all contents of the manuscript appropriately. The reviewer found the significant differences between tittle and contents of the manuscript. As the journal Micromachines focuses on micro-scaled device (i.e., organ-on-a-chip), the manuscript should be rewritten to focus on in-vitro model while minimizing the contents of physiology parts which should be required to understand 2D or 3D in-vitro model.

[2] While discussed and compared with the previous works, it would be better to summary important figures in a specific category of author’s decision. As most previous works were cited and discussed with only text even without representing vital figures, the manuscript made it difficult for audience to understand the manuscript. As a recommendation, vital figures of references were summarize as figures or tables properly.

[3] The manuscript did not include “CONCLUSION”. Additionally, “Outlook section (i.e., technical limitations, challenges)” should be included before “CONCLUSION”.

[4] In figure caption, there was no citation of the reference. Please add it while referring to previous papers.

Author Response

REVIEWER #1

General Comments

The manuscript reviewed lymphatic and blood vasculature systems in a point of physiology and in-vitro model. It will be taken attention to the audience of the micromachines. However, following issues the reviewer raised should be discussed before publication.

Response:

We thank the reviewer for the insightful comments and suggestions enumerated below in the specific concerns section. We have made substantial revisions including the creation of new figures and the review of more in vitro models to address some of the concerns and discussed the issues raised point-by-point below.

Specific comments include:

The manuscript focused on physiology (or biology) explanation instead of in-vitro model. Tittle did not present all contents of the manuscript appropriately. The reviewer found the significant differences between tittle and contents of the manuscript. As the journal Micromachines focuses on micro-scaled device (i.e., organ-on-a-chip), the manuscript should be rewritten to focus on in-vitro model while minimizing the contents of physiology parts which should be required to understand 2D or 3D in-vitro model.

Response:

We thank the reviewer for these comments, which provide an outside perspective on the content and structure of our review article. In attempt to better match our title to the content of the review, we reduced the amount of physiological content present in the review, specifically in the gut/intestine focused section. Additionally, we included information on ten more in vitro chip models to shift our focus more towards the micro-scaled devices and specifically highlighted how these devices mimicked the physiology upon which we elaborated in the sections not specifically focuses on in vitro models.

While discussed and compared with the previous works, it would be better to summary important figures in a specific category of author’s decision. As most previous works were cited and discussed with only text even without representing vital figures, the manuscript made it difficult for audience to understand the manuscript. As a recommendation, vital figures of references were summarize as figures or tables properly.

Response:

To accommodate this very helpful feedback, we have included two new figures (now Figures 2 and 3) and one new table (Table 1) to convey our information in a more visual manner. Figure 2 contains both immunostained tissue micrographs and chip schematics illustrating the techniques used by five different research labs in recapitulating blood vessels using in vitro models. Figure 3 also contains schematics and fluorescent images that demonstrate various chip model techniques, but specifically for lymphatic vessels on-chip. The images in both of these figures highlight the physiological aspects important to creating reliable and useful vasculature models, complementing the mixture of biological and modeling information present in our review. The table was included as a more-visual and less-text-heavy way to compare the advantages of different in vitro techniques, including both 2D and 3D modeling designs.

The manuscript did not include “CONCLUSION”. Additionally, “Outlook section (i.e., technical limitations, challenges)” should be included before “CONCLUSION”.

Response:

To make the manuscript more complete, we included a brief “Conclusion” section (now section 6) that summarizes our review. Within this conclusion, we also included the technical limitations and challenges of the field with a look towards the future. The serves as our “Outlook” section without specifically being labeled as an outlook. Our conclusion is also paired with the Table 1 mentioned in the previous specific concern response. It serves as a conclusion and review of the advantages and applications of several in vitro model techniques mentioned in our paper.

4) In figure caption, there was no citation of the reference. Please add it while referring to previous papers.

Responses:

We truly appreciate the reviewer catching this oversight on our end and have made sure to cite all references used for the figures in their captions.

Author Response

REVIEWER #2:

General Comments:

This manuscript by Henderson et al. reviews blood and lymphatic vasculatures on-chip platforms for applications in organ- and disease-specific in vitro modeling. Although the topic is interesting, this article mostly focused on the biology of blood and lymphatic vasculatures (development, heterogeneity, structure, functions, related diseases…) but not on the microfluidic devices (in vitro, on-chip platforms). This reviewer recommends the authors to shorten the biological part as well as extend the on-chip-related part of this manuscript to make it more suitable for readers of Micromachines. Several other comments are as follows.

Response:

We thank this reviewer for their insightful comments and general suggestions, including the praise that we have written on an interesting topic. We have made substantial revisions including ones to address the specific enumerated concerns below, which we address point-by-point. As a response to the general feedback of too much focus on physiology and not enough of the on-chip models, we have added at least ten more on-chip model references to the paper to shift the focus more towards the microfluidic devices. In addition, we have substantially shortened the biology portion focused on the intestine, reducing the content from 1.5 sections to only half of a section. To better frame the physiology as an integral portion of this paper, we have also included two new figures that show specific on-chip designs and how these models recreate the critical structure and physiology we discuss in the paper. This way the biology content complements the microfluidic device content and explains the importance of the physiology in terms of the more technical in vitro model content.

Specific Concerns:

Please properly cite the references when mentioning certain experiments/designs/ideas in the manuscript. For example, there should be references in lines 594 ~ 594 and Figure 3

Response:

We truly appreciate the reviewer catching this oversight on our end and have made sure to cite all references used for the figures in their captions as well as all experiments/designs/ideas mentioned, including adding the citation for the line listed here.

Please mention all the sub-figures in the context. For example, Figure 3(d) was not mentioned in the manuscript.

Response:

We truly appreciate the reviewer catching this additional oversight on our end and have made sure to mention all sub-figures in the manuscript, including Figure 3d (which is now known as Figure 5d).

Please use identical labels for all sub-figures. For example, Figure 3B and Figure 3c were used in the manuscript (upper or lower cases?).

Response:

Again, we truly appreciate the reviewer catching this oversight on our end and have made sure to label all figures and sub-figures identically (lower case letters with lower case Roman numerals, where applicable).

Since this is a review article, it would be great if the authors could list the advantages of microfluidic chips (2D and 3D with flow) compared with traditional dish -based platforms.

Response:

To further list the advantages of microfluidic chips compared to both 2D and 3D platforms, including traditional dish-based platforms (like transwell devices and endothelial monolayers), we created Table 1. Table 1 compares the applications and advantages of both brain and gut in vitro models mentioned in the review. In addition, the new Figures 2 and 3 (further elaborated below), show the effects of flow on the 3D structure. These three new visuals together compare various model platforms, including traditional dish-based ones, and clearly highlight the advantages of 3D microfluidic chips.

The authors are encouraged to review more microfluidic platforms for blood and lymphatic vasculatures with their structures detailed in texts and figures. There were only three figures in this manuscript.

Response:

To accommodate this very helpful feedback, we have included two new figures (now Figures 2 and 3) and one new table (Table 1), for a new figure total of 6 excluding the table, to convey our information in a more visual manner. Figure 2 contains both immunostained tissue micrographs and chip schematics illustrating the techniques used by an additional five different research labs in recapitulating blood vessels using in vitro models. Figure 3 also contains schematics and fluorescent images that demonstrate various chip model techniques, but specifically for lymphatic vessels on-chip using three more chips. The images in both of these figures highlight the physiological aspects important to creating reliable and useful vasculature models, complementing the mixture of biological and modeling information present in our review. The table was included as a more-visual and less-text-heavy way to compare the advantages of different in vitro techniques, including both 2D and 3D modeling designs. Beyond the new figures, additional chips were added to the brain and gut chip portions.

Please add a “Conclusion” section to make this article more complete.

Response:

To make the manuscript more complete, we included a brief “Conclusion” section (now section 6) that summarizes our review. Within this conclusion, we also included the technical limitations and challenges of the field with a look towards the future. The serves as our “Outlook” section without specifically being labeled as an outlook. Our conclusion is also paired with the Table 1 mentioned in the previous specific concern response. It serves as a more visual conclusion and review of the advantages and applications of several in vitro model techniques mentioned in our paper.

Round 2

Reviewer 1 Report

As the authors discussed appropriately the issues which the reviewer raised,

I recommend publication of the manuscript as current form.

Reviewer 2 Report

The authors have addressed all of my concerns. I am satisfied with this manuscript and recommending its publication in Micromachines in its present form.